# HoLoRA–Combining Orthogonal Fine-Tuning and LoRA with Householder Reflectors

## Abstract

The need for parameter-efficient fine-tuning (PEFT) has emerged as large pre-trained models are increasingly employed in specialized downstream tasks. Among PEFT methods, Low-Rank Adaptation (LoRA) is widely adopted due to its ability to fine-tune models with minimal additional parameters. However, LoRA's down-projection mechanism can lead to significant feature loss, particularly for tasks involving complex features and reasoning. This limitation poses a challenge in maintaining model performance in scenarios requiring high-dimensional representations. To address this issue, we introduce Householder Orthogonal LoRA (HoLoRA), which reparametrizes the down-projection matrix as a semi-orthogonal matrix, thereby mitigating feature loss. Our approach ensures strict orthogonality without increasing computational costs or modifying LoRA's core components. Experimental results on the GLUE benchmark show that HoLoRA consistently outperforms standard LoRA across various tasks, particularly in low-rank settings. By preserving essential features and improving fine-tuning efficiency, HoLoRA provides a robust solution to the limitations of existing PEFT methods. This advancement enhances LoRA's applicability in complex learning environments, promoting better performance in both low-budget and high-complexity scenarios.

## 1 Introduction

Recent research shows that models pre-trained on large datasets effectively capture general features and patterns within a data domain, exhibiting strong generalization and adaptation capabilities. This makes them highly suitable foundations for adaptation to specific downstream tasks. Fine-tuning large pre-trained models, such as Llama 2 (Touvron et al., 2023), with task-specific training data has become a mainstream approach for solving these downstream tasks (Shen et al., 2021). Compared to training models from scratch, fine-tuning leverages the knowledge learned by pre-trained models, reducing both time and computational costs while achieving improved performance.

However, driven by the scaling laws (Kaplan et al., 2020; Zhang et al., 2024a), the size of pre-trained models has increased rapidly. As a result, the cost for full-tuning (Raffel et al., 2020), which involves updating all the model parameters, has become extremely high for large-scale models. Full-tuning requires substantial computational and memory resources, as the tuning process requires the storage of all model parameters and their gradients.

To reduce the cost of fine-tuning, parameter-efficient fine-tuning (PEFT) methods have been introduced, which significantly decrease training costs while maintaining performance comparable to full-tuning. There are two main strategies in PEFT. The first is selective methods (Guo et al., 2020; Mahabadi et al., 2021), which involve choosing a subset of the original model parameters for tuning. However, these methods require the new task to be closely related to the one for which the model was pre-trained, making them less effective for tasks that are drastically different or involve cross-domain scenarios. The second strategy is additive methods, such as Adapters (Houlsby et al., 2019; Pfeiffer et al., 2020; Rücklé et al., 2020), Low-Rank Adaptation (LoRA) (Hu et al., 2022), and prefix-tuning (Li & Liang, 2021). These approaches add additional trainable components to the model's architecture and keep the original model frozen during training, allowing for efficient adaptation to new tasks while preserving the performance of the original model.

Most additive methods tend to modify the model architecture, which can result in an increased inference cost. However, LoRA is one of the most widely adopted methods because, after merging the additive component with the original model, it does not alter the model architecture. Typically, LoRA only tunes less than 1% of the model parameters, while achieving performance that is comparable to or even better than full-tuning.

In LoRA, given a pre-trained weight matrix $W_0$, the incremental update $\Delta$ is parameterized as the product of two low-rank matrices:

$$W = W_0 + \Delta = W_0 + BA^T, \tag{1}$$

where $W, W_0 \in \mathbb{R}^{m \times n}$, with $m$ and $n$ being the output and input dimensions of the layer, respectively. The matrices $B \in \mathbb{R}^{m \times r}$ and $A \in \mathbb{R}^{n \times r}$ have a rank $r$ such that $r \ll min\{m, n\}$. During training, the pre-trained weight $W_0$ is frozen and only the incremental matrices $A$ and $B$ are updated.

However, LoRA tends to perform poorly on complex tasks, particularly in multi-task learning and high-dimensional tasks such as mathematical reasoning and coding, which demand high representational power or reasoning capabilities (Biderman et al., 2024; Xin et al., 2024). If we regard matrices $A$ and $B$ as two consecutive layers without an activation function, the input layer $A$ acts as a highly down-projection matrix (as $r \ll min\{m, n\}$) which performs a feature compression of the input data. Such compression extracts important features for task-specific learning, but also poses the risk of feature loss. Furthermore, Zhang et al. (2024b; 2023b) also suggest that LoRA adaptation tends to be dominated by top singular vectors of the weight matrix, implying that the effective hidden dimensionality may be even smaller than the preset rank $r$. This limitation restricts LoRA's ability to capture the diverse and complex features required for more challenging tasks.

Orthogonality restriction can be used to alleviate the problem of feature loss for general neural network training (Ranasinghe et al., 2021). Considering both efficiency and budget constraints, research on LoRA variations (Zhang et al., 2023b;a) has introduced the use of a regularization term:

$$Reg(A) = \lambda \|A^T A - I\|_F^2, \tag{2}$$

to restrict the matrix $A$ to be near (semi-)orthogonal[1]. This method poses two potential challenges. Firstly, the calculation of the regularization term introduces additional computational steps, reducing training efficiency (Mao et al., 2024). Secondly, the introduction of a new hyperparameter $\lambda$ complicates tuning: a large $\lambda$ can slow convergence, while a small $\lambda$ may lead to ineffective orthogonal restriction. This raises the question that we address in this paper:

**Can we preserve the strict orthogonality of the down-projection matrix $A$ in LoRA during parameter updates without adding significant computational burden or introducing new hyperparameters?**

In response, we propose *HoLoRA*(Householder orthogonal Low-Rank Adaptation), which constructs an easy-to-compute function $h : \mathbb{L}^{n \times r} \to \mathbb{V}_{n,r}$, where $\mathbb{L}^{n \times r}$ is the space of $n \times r$ (non-strict) lower triangular matrices, and $\mathbb{V}_{n,r}$ is the Stiefel manifold (the set of $n \times r$ semi-orthogonal matrices). We reparametrize the matrix $A$ as:

$$A = h(M_A) \tag{3}$$

and only learn the parameters in $M_A$, so that A can be kept orthogonal during the whole training process. Compared to other LoRA variation methods, our main contributions are:

1. **Strict orthogonality:** We impose a strict orthogonal restriction on the given matrices, which better alleviates the problem of feature loss.

2. **Computation efficiency:** The function $h$ is easy to compute, so the training time is similar as vanilla LoRA.

3. **Compatibility with LoRA variants:** Our method preserves the fundamental structure of vanilla LoRA, ensuring compatibility with many other LoRA variation techniques.

---

[1]For simplicity, we refer to both orthogonal and semi-orthogonal matrices as orthogonal, where a semi-orthogonal matrix is a non-square matrix with orthonormal columns that preserves most properties of orthogonal matrices

## 2 PRELIMINARIES AND RELATED WORKS

### 2.1 LoRA

LoRA (Hu et al., 2022) is inspired by the finding that the incremental update weight matrix of fine-tuning has intrinsic dimensions that can be approximated by low-rank matrices. LoRA parameterizes the update matrix $\Delta W$ with two learnable matrices, significantly reducing the number of trainable parameters. The formula of the LoRA weight update for a weight matrix in one single layer is:

$$W = W_0 + \Delta W = W_0 + \frac{\alpha}{r}BA^T, \tag{4}$$

where $W_0, W \in \mathbb{R}^{m \times n}$ are the weight matrices before and after the update. $\Delta W \in \mathbb{R}^{m \times n}$ is the incremental update matrix, referred to as the "delta adaptation block". The matrices $B \in \mathbb{R}^{m \times r}$ and $A \in \mathbb{R}^{n \times r}$ contain the learnable parameters, and $m, n$ are the output and input dimensions, respectively. The scalar $\alpha$ is a hyperparameter controlling the scaling factor of the delta block[2]. $r$ is the rank of delta block, which is the preset budget parameter, as the total learnable parameter of a weight matrix will be $(m + n) \cdot r$. Given the following property:

$$\text{rank}(BA^T) \leq \min\{\text{rank}(A), \text{rank}(B)\}, \tag{5}$$

the delta block is low rank and upper-bounded by $r$. Extensions to LoRA, such as AdaLoRA (Zhang et al., 2023b), explore adaptive budget allocation for PEFT, improving performance by dynamically reallocating rank budgets among layers according to their importance.

Other approaches, such as IncreLoRA (Zhang et al., 2023a), suggest incremental parameter allocation during training to better utilize the model's capacity and avoid underfitting of critical layers, while DyLoRA (Valipour et al., 2022) dynamically adjusts the low-rank adaptation throughout training to improve generalization on various downstream tasks. These extensions address limitations of static budget allocation in LoRA and provide pathways for more efficient fine-tuning of large language models. However, none of these methods ensure strict orthogonality within the learned parameters, which can lead to feature loss in some complex tasks.

### 2.2 ORTHOGONAL FINE-TUNING (OFT)

Orthogonal Fine-Tuning (OFT) (Qiu et al., 2023; Yuan et al., 2024) adapts the pre-trained model weights by multiplying them with an orthogonal matrix:

$$W = UW_0, \tag{6}$$

where $U$ is an orthogonal matrix. This orthogonality constraint is intended to mitigate feature loss during adaptation by preserving important properties of the pre-trained weights. Regularization methods, as seen in Equation (2), have been explored to maintain the orthogonality of $U$. However, balancing computational efficiency with strict orthogonality remains a challenge.

The study by (Yuan et al., 2024) proposes an orthogonality restriction for OFT using Householder reflectors, offering a way to impose strict orthogonality without significant computational overhead. However, their approach is limited to square matrices and does not directly apply to LoRA's down-projection matrices, which are typically rectangular.

OLoRA (Orthonormal LoRA) (Büyükakyüz, 2024) aims to address feature preservation by encouraging orthogonality within the low-rank adaptation. However, this is achieved via regularization, which introduces additional computational costs and requires tuning of hyperparameters, such as the regularization strength $\lambda$. Thus, this approach struggles to achieve a balance between computational efficiency and strict orthogonality.

In contrast to these existing methods, our proposed HoLoRA directly addresses the issue of maintaining strict orthogonality in the low-rank adaptation matrices without introducing significant computational overhead or additional hyperparameters. By reparametrizing the matrix $A$ using Householder transformations, HoLoRA ensures that the down-projection matrix remains orthogonal throughout training. This strict orthogonality better preserves feature representation, mitigating the risk of feature loss without the need for regularization terms or tuning of hyperparameters like $\lambda$, which are required in other approaches.

---

[2]For convenience, we omit the scaling factor $\frac{\alpha}{r}$ in later discussion, as this factor is introduced primarily for learning rate scheduling (Hu et al., 2022).

## 3 METHOD

### 3.1 PRELIMINARY ON HOUSEHOLDER REFLECTORS

We first introduce the definition of a Householder reflector (Householder, 1958; Mhammedi et al., 2017):

**Definition 1** *For any $n, k \in \mathbb{N}$ with $k \leq n$, and any vector $u \in \mathbb{R}^k$, the $n$-dimensional Householder reflector parametrized by $u$ is defined as:*

$$
H_k^n(u) = \begin{cases} \begin{pmatrix} I_{n-k} & 0 \\ 0 & I_k - \frac{2uu^T}{\|u\|^2} \end{pmatrix} & \text{if } u \neq \mathbf{0}, \\ I_n & \text{if } u = \mathbf{0}. \end{cases} \tag{7}
$$

*We refer to the vector $u$ as the **Householder vector**.*

**Remark 1** *Specifically, when $k = n$, we denote the Householder reflector simply as $H(u)$. A Householder reflector performs a reflection in the $n$-dimensional space, with the reflection plane determined by the vector $u$.*

*Each Householder reflector with $k = n$ can be expressed as:*

$$
H_k^n(u) = H(p(u)) = H(\tilde{u}) = I_n - 2\tilde{u}\tilde{u}^T, \tag{8}
$$

*where $p(u)$ is an $n$-dimensional vector obtained by padding the first entries of $u$ with zeros. The vector $\tilde{u}$ is the normalized form of $p(u)$, with unit norm:*

$$
\tilde{u} = \frac{p(u)}{\|p(u)\|}. \tag{9}
$$

*This normalization ensures that the reflection is properly defined, maintaining the orthogonal properties of the Householder reflector.*

**Remark 2** *The Householder reflector is determined solely by the direction of the Householder vector $u$. Therefore, scaling $u$ by any non-zero coefficient does not alter the Householder reflector $H_k^n(u)$.*

For simplicity, and since the probability of the vector $u$ becoming zero during training is negligible when initialized with non-zero values, we assume $u \neq \mathbf{0}$ throughout our discussion. This ensures that the Householder reflector function $H_k^n(u)$ is continuous with respect to $u$.

There are some important properties of the Householder reflector:

**Theorem 1** *The Householder reflector is an orthogonal matrix for any arbitrary vector $u$ (Householder, 1958):*

$$
(H_k^n(u))^T H_k^n(u) = I_n. \tag{10}
$$

Moreover, any square orthogonal matrix can be represented as a product of Householder reflectors (Mhammedi et al., 2017):

**Lemma 1 (Householder decomposition)** *For any $n \times n$ orthogonal matrix $U$, there exist $n$ Householder reflectors $H_n^n(u_n), H_{n-1}^n(u_{n-1}), \ldots, H_1^n(u_1)$, where $u_i \in \mathbb{R}^i$ for $i = 1, 2, \ldots, n$, such that*

$$
U = H_n^n(u_n)H_{n-1}^n(u_{n-1}) \cdots H_1^n(u_1). \tag{11}
$$

**Remark 3** *As per the definition of the Householder reflector, the upper-left part of a Householder reflector is an identity matrix. Therefore, the first $r$ columns of $U$ are uniquely determined by the product $H_n^n(u_n)H_{n-1}^n(u_{n-1}) \cdots H_{n-r+1}^n(u_{n-r+1})$. Multiplying by the remaining reflectors $H_{n-r}^n(u_{n-r}), \ldots, H_1^n(u_1)$ on the right does not alter the first $r$ columns.*

Although a semi-orthogonal matrix of size $n \times r$ can be derived by slicing the first $r$ columns of $U$, computing the product of many consecutive $n \times n$ matrices can be computationally expensive. To address this, we adopt the Truncated CWY (T-CWY) representation (Likhosherstov et al., 2021) form of Householder reflectors in our method.

### 3.2 TRUNCATED CWY FORM OF HOUSEHOLDER REFLECTORS

The T-CWY parametrization combines the product of multiple Householder reflectors into a fixed-length matrix multiplication formula, thereby reducing computational and memory costs. To introduce T-CWY, we first present the CWY parametrization form (Likhosherstov et al., 2021):

**Theorem 2 (CWY Parametrization)** *For any set of nonzero $N$-dimensional real-valued vectors $\{u^{(i)}\}_{i=1}^{L}$, we have*

$$H(u^{(1)})H(u^{(2)}) \cdots H(u^{(L)}) = I_N - MK(M)^{-1}M^T, \tag{12}$$

*where*

$$M = \left[ \frac{u^{(1)}}{\|u^{(1)}\|_2}, \frac{u^{(2)}}{\|u^{(2)}\|_2}, \ldots, \frac{u^{(L)}}{\|u^{(L)}\|_2} \right] \in \mathbb{R}^{N \times L},$$

*and*

$$K(M) = \frac{1}{2}I_L + \mathrm{striu}(M^T M) \in \mathbb{R}^{L \times L}.$$

*Here,* $\mathrm{striu}(\cdot)$ *denotes the function that returns the argument matrix with all diagonal and lower-triangular elements zeroed out.*

T-CWY parametrization provides an efficient method to parametrize the first $r$ columns of the expression in formula (12).

**Theorem 3 (T-CWY Parametrization)** *Suppose an $N \times N$ orthogonal matrix $U$ has a House-holder decomposition given by*

$$U = H(u^{(1)})H(u^{(2)}) \cdots H(u^{(L)}). \tag{13}$$

*Then, the first $r$ columns of $U$ can be represented as*

$$U_{[:,:r]} = I_{N[:,:r]} - MK(M)^{-1}(M_{[:r,:]})^T, \tag{14}$$

*where $M$ and $K$ are defined as in Theorem 2, and the subscripts $[:,: r]$ and $[: r,:]$ indicate sub-matrices formed by slicing the first $r$ columns and rows, respectively.*

Note that in the context of LoRA, $r$ is much smaller than the model dimension $N$, implying $r \ll N$. Consequently, the computational complexity of inverting the $r \times r$ matrix $K(M)$ is relatively low, making the T-CWY parametrization efficient.

### 3.3 HOLORA METHOD

Considering Lemma 1 and Theorem 3, we naturally arrive at an orthogonal parametrization of the down-projection matrix $A$ in the LoRA equation (1) as follows:

$$A = h(M_A), \quad M_A \in \mathbb{L}^{n \times r}, \tag{15}$$

$$h(M_A) = I_{N[:,:r]} - \tilde{M}_A K(\tilde{M}_A)^{-1} \tilde{M}_{A[:r,:]}^T, \tag{16}$$

$$W = W_0 + Bh(M_A)^T, \tag{17}$$

where the learnable parameters are $B$ and $M_A$. The total number of learnable parameters is $(r-1)^2$ fewer than in vanilla LoRA, as $M_A$ is constrained to be lower triangular.

Here, $\tilde{M}_A$ is derived by normalizing the columns of $M_A$, analogous to the normalization of $M$ in Theorem 2. The notation $\mathbb{L}^{n \times r}$ represents the space of $n \times r$ (non-strict) lower triangular matrices. We restrict the domain of $M_A$ to be lower triangular to enable an equivalent T-CWY parametrization, as in Lemma 1, where each vector $u_i \in \mathbb{R}^i$.

Figure 1: HoLoRA structure and our reparametrization of down-projection matrix. Blue indicates static (frozen) parameters, red indicates learnable parameters, and green indicates dependent parameters derived from other learnable/frozen parameters

### 3.3.1 INITIALIZATION OF HOLORA

For the matrix $B$, we use the same zero initialization as in LoRA, ensuring that the adaptation block is initialized to zero. In contrast, vanilla LoRA initializes the matrix $A$ using a uniform distribution. However, empirical experiments have shown that initializing each entry of $M_A$ from the same distribution results in $h(M_A)$ being extremely close to the (generalized non-square) identity matrix.

To address this, we aim to initialize $M_A$ such that $h(M_A)$ is uniformly sampled from the Stiefel manifold $\mathbb{V}_{n,r}$. For this purpose, we utilize the following theorem from (Nirwan & Bertschinger, 2019):

**Theorem 4** *If the vectors $\{v_i\}_{i=1}^n$ are uniformly distributed on the unit sphere in $\mathbb{R}^i$, and we set*

$$u_i = \frac{v_i + \text{sgn}(v_{i1})\|v_i\|e_1}{\|v_i + \text{sgn}(v_{i1})\|v_i\|e_1\|},$$

*and define the Householder reflector as*

$$\tilde{H}_i^n(u_i) = -\text{sgn}(v_{i1})H_i^n(u_i),$$

*then the matrix*

$$U = \tilde{H}_n^n(u_n)\tilde{H}_{n-1}^n(u_{n-1})\cdots\tilde{H}_1^n(u_1) \tag{18}$$

*is uniformly distributed according to the Haar measure in $O_n(D)$, where $O_n(D)$ is the group of $n \times n$ orthogonal matrices.*

Considering Remark 3, to achieve a uniform distribution of $h(M_A)$ on the Stiefel manifold $\mathbb{V}_{n,r}$, we generate the vectors $v_n, v_{n-1}, \ldots, v_{n-r+1}$ uniformly on their respective unit spheres. If $v_{i1} > 0$, we multiply $v_i$ by $-1$ to ensure the correct orientation. We then calculate the corresponding vectors $u_n, u_{n-1}, \ldots, u_{n-r+1}$ as per Theorem 4. The last $(n - i + 1)$ entries of the $i$-th column of $M_A$ are set to be $u_{n-i+1}$ for $i = 1, \ldots, r$. This procedure ensures that the generated $h(M_A)$ has a uniform distribution on the Stiefel manifold $\mathbb{V}_{n,r}$.

### 3.3.2 TIME COMPLEXITY OF HOLORA

In this section, we demonstrate that the time complexity added by computing $h(M_A)$ is negligible compared to the vanilla LoRA method.

The computation of the function $K$ has a time complexity of $O(nr^2)$. Since $K$ is an $r \times r$ matrix, its inversion requires a time complexity of $O(r^3)$. The matrix operations involved in equation (16) also have a time complexity of $O(nr^2)$, while the normalization of $M_A$ takes time $O(nr)$. Therefore, the overall time complexity for calculating the function $h$ is $O(nr^2)$.

In contrast, vanilla LoRA involves the multiplication of an $m \times r$ matrix with an $r \times n$ matrix, which has a time complexity of $O(mnr)$.

Given the assumption that $r \ll \min\{m, n\}$, the time complexity of computing $h(M_A)$ is negligible when compared to the time complexity of vanilla LoRA.

---

**Algorithm 1** HoLoRA - Householder Orthogonal Low-Rank Adaptation

---

1: **Input:** Pretrained weights $W_0 \in \mathbb{R}^{m \times n}$), low-rank factor $r$, training data $D$, learning rate $\eta$, number of epochs $E$
2: **Output:** Adapted matrices $B \in \mathbb{R}^{m \times r}$ and $M_A \in \mathbb{L}^{n \times r}$
3: Initialize $B$ as zero matrix and $M_A$ as in Theorem 4.
4: Keep the original pretrained weights $W_0$ frozen
5: **for** epoch $e = 1$ to $E$ **do**
6:     **for** each data batch $(x, y)$ in $D$ **do**
7:         Compute adapted weights: $W' = W_0 + Bh(M_A)^\top$
8:         Forward pass: compute prediction $\hat{y}$ using $W'$
9:         Compute loss $L(\hat{y}, y)$
10:        Backpropagate and update $B$: $B \leftarrow B - \eta \nabla_B L$
11:        Backpropagate and update $M_A$: $M_A \leftarrow M_A - \eta \nabla_{M_A} L$
12:     **end for**
13: **end for**
14: **Return:** Final adapted matrices $B$ and $M_A$

---

### 3.4 COMPATIBILITY OF HoLoRA WITH OTHER LoRA VARIATIONS

Since HoLoRA only changes the parametrization of the down-projection layer in LoRA, without altering any structural or learning schedule aspects, it remains compatible with a wide range of LoRA variations. We provide insights into possible combinations for future research.

#### 3.4.1 HoLoRA WITH SVD DYNAMIC RANK ALLOCATION

SVD (Singular Value Decomposition) dynamic rank allocation methods (Zhang et al., 2023b;a) for LoRA parametrize the adaptation block in a truncated SVD-like form rather than using $A$ and $B$ matrices:

$$\Delta W = USV^T, \tag{19}$$

where $U \in \mathbb{R}^{m \times r}$, $V \in \mathbb{R}^{n \times r}$, and $S \in \mathbb{R}^{r \times r}$.
The SVD form requires $U$ and $V$ to be orthogonal matrices and $S$ to be diagonal. Current methods such as AdaLoRA (Zhang et al., 2023b) and IncreLoRA (Zhang et al., 2023a) enforce these conditions using regularization as described in equation (2). Alternatively, the orthogonal restriction can be achieved by utilizing the function $h$ in HoLoRA.

By Remark 3, adding or removing the last columns of $M_A$ does not affect the first columns of $h(M_A)$. Therefore, during dynamic rank allocation, we can adjust the rank by adding or deleting columns of $M_A$, ensuring that the retained singular vectors remain unchanged.

#### 3.4.2 HoLoRA WITH LoRA INITIALIZATION

OLoRA (Büyükakyüz, 2024) leverages orthonormal matrix initialization via QR decomposition to accelerate LoRA convergence. Since Householder reflectors were originally designed for performing QR decomposition (Householder, 1958), we can use the Householder QR decomposition method to easily obtain the Householder vectors. These vectors can then be used to initialize the columns of matrix $M_A$ in HoLoRA, achieving an equivalent initialization to OLoRA.

LoRA-XS (Bałazy et al., 2024) computes the SVD of the pre-trained weight matrix and initializes the LoRA delta block using information derived from the SVD form. If we employ the Golub-Reinsch SVD algorithm (Golub & Kahan, 1965)—a method that uses Householder reflectors and is still widely used for its efficiency—the parameters for the Householder reflectors can be obtained and subsequently used in HoLoRA during training.

# 4 EXPERIMENTS

We fine-tune DeBERTaV3-base (He et al., 2021) with our proposed HoLoRA method, on 8 natural language understanding tasks from General Language Understanding Evaluation (GLUE) (Wang, 2018) benchmark. We compare scores for each task and the total average score.

## 4.1 EXPERIMENT SETTINGS

**Dataset** The GLUE benchmark (Wang, 2018) is widely used to evaluate the generalization ability of models across various natural language understanding tasks. We choose eight tasks (MNLI, SST-2, CoLA, QQP, QNLI, RTE, MRPC, STS-B) from GLUE for a general evaluation on single sentence tasks, similarity and paraphrase tasks, and inference tasks. For the evaluation metrics, We use the overall accuracy for MNLI, Matthew's correlation for CoLA, Pearson correlation for STS-B, and accuracy for all other tasks.

**Baselines**

We compare our method with full-tuning and vanilla LoRA (Hu et al., 2022). LoRA generally performs better than other PEFT methods such as Adapters (Houlsby et al., 2019; Pfeiffer et al., 2020). We focus on the comparison to LoRA as our method is a general approach to mitigate LoRA's limitation of feature loss. HoLoRA remains compatible with newer LoRA variants.

**Implementation details** To conduct a fair comparison, we build on the codebase of LoRA (Hu et al., 2022). We use the pre-trained DeBERTaV3-base (He et al., 2021) model from HuggingFace, which contains 12 Transformer encoder layers with 768 hidden units, and has a total parameter size of 86M. Similar to the experimental settings used in the LoRA paper, we adapt only the query (Q) and value (V) layers of the pre-trained model. For a comparison under different fine-tuning parameter budgets, we perform two sets of experiments, with the rank $r$ set to be 2 and 8 respectively. For MRPC, RTE, and STS-B tasks, we start from the LoRA-adapted MNLI checkpoint for a same setting as LoRA paper. We use the same hyperparameter settings in Table 3, referring to LoRA paper (Hu et al., 2022) and AdaLoRA paper Zhang et al. (2023b)[3], for both LoRA and HoLoRA. All experiments are conducted on NVIDIA L4 Tensor Core GPUs through Google Cloud Platform.

## 4.2 EXPERIMENT RESULTS

The results of adapting DeBERTaV3-base model using Full-Tuning, LoRA and HoLoRA across eight different tasks under budget level $r = 2$ and $r = 8$ are shown in table 1. HoLoRA achieves better performance compared with baseline methods in most tasks, especially under low budget level. It is well noted that in SST-2, CoLA, RTE tasks, HoLoRA with low budget level ($r = 2$) outperforms LoRA with 4 times of budget level ($r = 8$). Overall, the average score for HoLoRA under low budget level (88.07) is higher than the score of LoRA under high budget level (87.59). The results show that HoLoRA can mitigate the feature loss problem in LoRA, which is more prominent under low budget level settings.

## 4.3 ORTHOGONALITY CHECK OF VANILLA LoRA

In the experiment of vanilla LoRA, we also check the orthogonality of matrix A to see if it is near orthogonal. We use the evaluation metric

$$Ort(A) = ||\tilde{A}^T \tilde{A} - I||_F \tag{20}$$

---

[3]AdaLoRA paper conducts LoRA experiments adapting all the Q,K,V, and FFN layers of DeBERTaV3-base model, which is a good reference as we want to tune the Q and V layers of DeBERTaV3-base model. Experiments in LoRA (Hu et al., 2022) do not use the DeBERTaV3-base model

Table 1: Experiment results of fine-tuning DeBERTaV3-base on GLUE benchmark. Evaluation metrics: Overall (matched and mismatched) accuracy for MNLI, Matthew's correlation for CoLA, Pearson correlation for STS-B, and accuracy for other tasks.

| | #Params | MNLI Acc. | SST-2 Acc. | CoLA Mcc. | QQP Acc. | QNLI Acc. | RTE Acc. | MRPC Acc. | STS-B Corr. | ALL Avg. |
|---|---|---|---|---|---|---|---|---|---|---|
| Full-Tuning | 86M | **90.55** | 89.53 | 68.50 | 91.35 | 93.30 | 75.81 | 87.99 | 89.53 | 85.82 |
| LoRA$_{r=2}$ | 0.1M | 89.59 | 94.72 | 64.25 | **90.56** | **93.94** | 84.48 | **89.95** | **91.90** | 87.42 |
| HoLoRA$_{r=2}$ | 0.1M | **89.90** | **94.84** | **69.57** | 90.20 | 93.90 | **85.56** | 88.97 | 91.59 | **88.07** |
| LoRA$_{r=8}$ | 0.3M | **90.20** | 94.72 | 66.31 | **91.37** | **94.07** | 83.75 | 88.73 | 91.53 | 87.59 |
| HoLoRA$_{r=8}$ | 0.3M | 89.85 | **95.64** | 68.10 | 90.48 | 93.79 | 83.75 | **90.44** | **91.68** | **87.97** |

to measure the degree of mutual orthogonality between the columns of matrix $A$, where $\tilde{A}$ is obtained by normalizing each column of A.

We plot Figure 2 for the average value of $Ort(A)$ in all adaptation layers versus number of iterations. It can be seen that the value of $Ort(A)$ increase rapidly during early stages, and then decrease slowly. However, the minimum value is still much larger than the threshold $10^{-3}$, below which we regard a matrix to be near-orthogonal (Zhang et al., 2023b), so the matrix $A$ in LoRA is far from orthogonal during training.

The orthogonality check indicates that the LoRA block $A$ will not converge to be orthogonal without Householder orthogonal restriction, and HoLoRA learns a different adaptation block compared with LoRA.

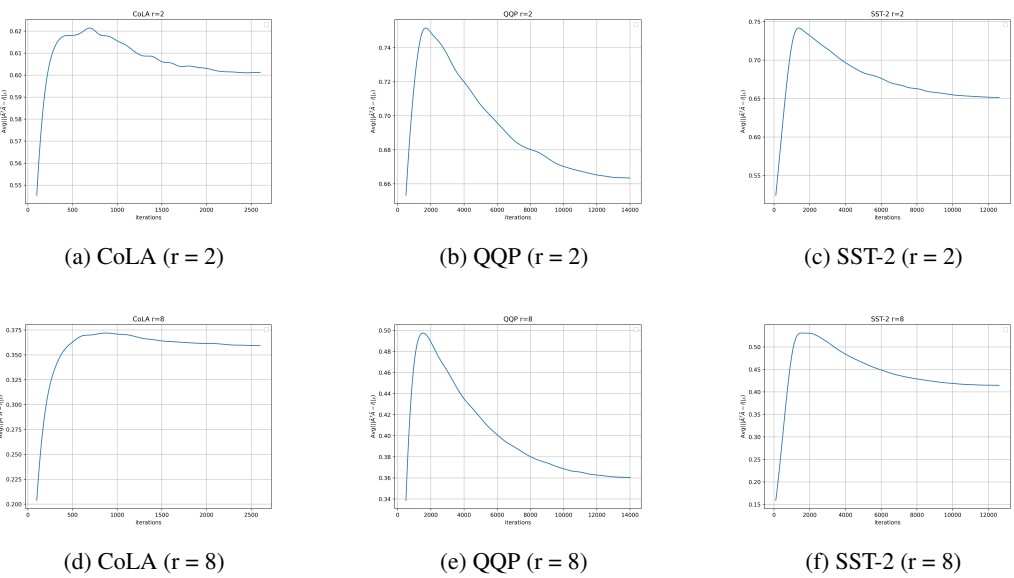

(a) CoLA (r = 2)  (b) QQP (r = 2)  (c) SST-2 (r = 2)

(d) CoLA (r = 8)  (e) QQP (r = 8)  (f) SST-2 (r = 8)

Figure 2: Average orthogonality of layer A in LoRA during training. With x-axis iterations and y-axis $Ort(A)$. Lower values means more near-orthogonal.

### 4.4 EMPIRICAL TIME COMPARISON

We also present the actual time cost comparison of LoRA and HoLoRA in Table 2 to assess the feasibility of using HoLoRA as a drop-in replacement for LoRA. It can be seen that, for all three tasks, the additional time for HoLoRA is within 10% of LoRA. Meanwhile, for QQP task, which is more complex and takes more time, the additional training time cost is less than 3%. A possible reason is that the additional initialization time take a constant additional time for HoLoRA, so the

proportion of overall additional training time will be lower for complex tasks in which initialization time can be neglected.

Table 2: Comparison of Training Time cost (seconds) and the additional time (in proportion %) for LoRA and HoLoRA fine-tuning on QQP, RTE, and MRPC tasks.

| Task | Budget | LoRA (s) | HoLoRA (s) | Additional Time (%) |
|------|--------|----------|------------|---------------------|
| **QQP** | $r = 2$ | 22645 | 23039 | 1.74% |
| | $r = 8$ | 22631 | 23131 | 2.21% |
| **RTE** | $r = 2$ | 328 | 320 | -2.44% |
| | $r = 8$ | 315 | 319 | 1.27% |
| **MRPC** | $r = 2$ | 134 | 145 | 8.21% |
| | $r = 8$ | 136 | 149 | 9.56% |

## 5 CONCLUSION

We propose Householder Orthogonal Low-Rank Adaptation (HoLoRA) to address feature loss in LoRA by adding strict orthogonality restriction on the down-projection matrix. HoLoRA preserves important features while maintaining computational efficiency and compatibility with existing LoRA variations. We conduct experiments on the GLUE benchmark, showing that HoLoRA outperforms vanilla LoRA, particularly in low-budget scenarios where the risk of feature loss is highlighted. This work highlights the value of orthogonality in LoRA, providing insights for future research on integrating HoLoRA with other LoRA methods.

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

# A APPENDIX

Table 3: Hyperparameters for LoRA and HoLoRA on GLUE benchmark.

| Dataset | learning rate | batch size | epochs | $\alpha$ (r = 2 / r = 8) |
|---------|---------------|------------|--------|--------------------------|
| MNLI | $5.0 \times 10^{-4}$ | 32 | 7 | 8 / 16 |
| SST-2 | $8.0 \times 10^{-4}$ | 32 | 24 | 16 / 16 |
| CoLA | $8.0 \times 10^{-4}$ | 16 | 20 | 32 / 32 |
| QQP | $8.0 \times 10^{-4}$ | 32 | 5 | 8 / 16 |
| QNLI | $7.0 \times 10^{-4}$ | 32 | 4 | 32 / 32 |
| RTE | $1.2 \times 10^{-3}$ | 32 | 11 | 8 / 32 |
| MRPC | $1.0 \times 10^{-3}$ | 32 | 10 | 8 / 32 |
| STS-B | $2.0 \times 10^{-3}$ | 32 | 10 | 16 / 32 |

