# OpenReview forum: "HoLoRA: Combining Orthogonal Fine-Tuning and LoRA with Householder Reflectors"
_ICLR.cc/2025/Conference — ICLR 2025 Conference Withdrawn Submission_

### Official Review · Reviewer_iG6r · 2024-10-31

**Soundness:** 2
**Presentation:** 2
**Contribution:** 1
**Rating:** 3
**Confidence:** 5

**Summary:**

This paper proposes a novel method HoLoRA, aiming to make LoRA’s down projection matrix A orthogonal during training.

**Strengths:**

1. Good motivation.
2. Well Presentation.

**Weaknesses:**

1. All the theorems and lemmas do not have proofs.
2. As shown in Eqs. (15-17), HoLoRA also increases computational costs, which is contrary to the presentation in Line 20, “without increasing computational costs”.
3. First, the author employed a bunch of fancy mathematics to derive an orthogonal function in the end. However, these mathematical proofs were not originally proposed by the author; in fact, the author merely used an existing orthogonal tool. Second, to keep A orthogonal and without introducing a regularizer, why not using Cayley Parameterization? It is more easier and computation efficient. One can learn A, B and let LoRA be \delta W = B (I+A)(I−A)^{−1}. Third, there lacks ablation studies with other orthogonal methods like Cayley Parameterization.
4. The most critical issue with this paper is the lack of experiments. Currently, PEFT methods are typically tested on large models such as LLaMA-7B. Even taking into account the possibility that the authors may have limited GPU resources, the NLU experiments only include a comparison between two different parameter sizes. Moreover, considering that NLU experiments are susceptible to the influence of random seeds, it is unclear whether the results reported by the authors are the averages of multiple experimental runs.
5. Line 405-406, ”and has a total parameter size of 86M”.  The total parameter size of DeBERTaV3-base should be 184M.
6. The paper misses a subsection: an explanation of why the orthogonality constraint can mitigate feature loss. While the authors state that it can, they do not provide a clear rationale or reasoning behind this claim.
7. Regarding weakness 6, there is no experiment validating that HoLoRA can decrease the feature loss.

**Questions:**

1. Line 72-73. “Furthermore, Zhang et al. (2024b; 2023b) also suggest … the weight matrix.” Could you please explain how Zhang et al. 2023b (AdaLoRA) suggest that LoRA adaptation tends to be dominated by top singular vectors of the weight matrix?
2. B is another projection matrix. Why do not make B orthogonal?

---

### Official Review · Reviewer_ndX1 · 2024-11-02

**Soundness:** 3
**Presentation:** 3
**Contribution:** 3
**Rating:** 3
**Confidence:** 2

**Summary:**

This paper addresses the limitations of LoRA, specifically focusing on the feature loss that occurs during LoRA’s down-projection, which can impair model performance on complex, high-dimensional tasks. The authors propose Householder Orthogonal LoRA (HoLoRA), an approach that reparametrizes LoRA's down-projection matrix as a semi-orthogonal matrix to reduce feature loss, while preserving LoRA’s computational efficiency and structural simplicity. Experimental results on the GLUE benchmark validate HoLoRA’s efficacy, showing certain performance improvements over standard LoRA in low-rank settings.

**Strengths:**

1.	The authors study the timely issue of parameter-efficient finetuning.
2.	This paper considers the potential feature loss when utilizing LoRA, in particular when the effective hidden dimensionality may be even smaller than the preset rank r. As such, the authors aim to render the column vectors of each A-matrix to be orthogonal.

**Weaknesses:**

1.	Orthogonal fine-tuning (OFT) has been the subject of extensive research in recent years. Although this paper introduces a method based on the Householder reflector, its contributions appear to be closely aligned with existing studies, resulting in a novelty that is somewhat limited.
2.	A thorough comparison with previous OFT algorithms is warranted. Specifically, it remains unclear whether the proposed Householder method, which enforces hard orthogonality, demonstrates superiority over approaches that utilize regularization techniques for soft orthogonality. Additionally, the authors should discuss the computational and memory requirements associated with their method.
3.	The authors’ comparisons in Table 1 are primarily limited to classical methods. Recent advancements, such as DoRA, propose alternative strategies by decoupling the magnitude and direction of LoRA, suggesting that enforcing orthogonality is not the sole approach to enhancing overall performance.
4.	The comparison of the proposed algorithm to LoRA is restricted to experiments conducted on DeBERTa with relatively small datasets. It would strengthen the findings if the authors tested their approach on larger backbone models to evaluate its scalability and effectiveness.
5.	Several equations in the paper appear to contain inaccuracies. For instance, H(p(u)) is not equivalent to H(\tilde{u}) as stated in Equation (8). A thorough review of these equations is recommended to ensure correctness.
6.	The axis labels in Figure 2 are too small, making them difficult to read.

**Questions:**

see weakness

---

### Official Review · Reviewer_dSDt · 2024-11-03

**Soundness:** 3
**Presentation:** 3
**Contribution:** 1
**Rating:** 3
**Confidence:** 5

**Summary:**

This paper proposes Householder Reflectors to employ orthogonality constrains on LoRA parameters efficiently.

The paper is well written. In the experimental analyses, the proposed HoLoRA performs on par with the baseline LoRA.

**Strengths:**

In the experimental analyses, the proposed HoLoRA performs on par with the baseline LoRA, and outperforms the LoRA in some benchmark tasks.

**Weaknesses:**

Householders have been used in numerical computation of operations on orthogonal matrices. Therefore, the proposed method is a trivial application of Householders with orthogonal constraints. Therefore, the novelty of the method is limited.

In order to improve the novelty, the proposed HoLoRA should be compared with all the related work conceptually, theoretically and experimentally.

**Questions:**

- The proposed HoLoRA performs on par with LoRA on various benchmarks. That is, it either outperforms or underperforms LoRA with small margin. Could you elaborate these results?

- Following the previous question, could you please compare HoLoRA with the other methods that employ orthogonality constraints on LoRA matrices, especially to see whether the performance issue is related to failure of orthogonality constraints or the method?

- In numerical implementation of orthogonality constraints, algorithms implement Householders implicitly in the linear algebra operations, e.g. in Pytorch or Tensorflow. Then, what are the main differences between these numerical Householder calculations and the proposed operations?

- How does Householders perform in the other AI tasks, e.g. in text and image generation tasks?

---

### Official Review · Reviewer_13Uu · 2024-11-03

**Soundness:** 2
**Presentation:** 2
**Contribution:** 3
**Rating:** 3
**Confidence:** 3

**Summary:**

This article introduces a novel parameter-efficient fine-tuning method called HoLoRA (Householder Orthogonal Low-Rank Adaptation), which addresses the issue of feature loss when fine-tuning large pre-trained models for specific downstream tasks. HoLoRA mitigates feature loss by reparameterizing the down-projection matrix as a semi-orthogonal matrix, while maintaining computational efficiency and compatibility with existing LoRA variants. Experimental results on the GLUE benchmark for various natural language understanding tasks demonstrate that HoLoRA outperforms standard LoRA, especially under low-budget scenarios. By preserving critical features and enhancing fine-tuning efficiency, HoLoRA offers an effective solution to the limitations of current parameter-efficient fine-tuning methods, showcasing its potential to improve model performance in both low-budget and high-complexity scenarios.

**Strengths:**

1. The proposed HoLoRA enforces strict orthogonality in the down-projection matrix through efficient Householder transformations, minimizing feature loss without adding significant computational cost.
2. The proposed HoLoRA outperforms standard LoRA in low-rank scenarios, showing stronger feature preservation and improved accuracy on complex tasks, particularly with limited parameters.
3. The proposed method maintains LoRA’s core structure, allowing seamless integration with other LoRA variations and enabling flexible, parameter-efficient fine-tuning across diverse applications.

**Weaknesses:**

1. The experiments focus solely on the DeBERTaV3-base model, without testing on other pre-trained models (e.g., GPT, BERT, T5). This limited scope makes it difficult to assess HoLoRA's effectiveness across different model architectures.
2. The evaluation is based on the GLUE benchmark, primarily covering natural language understanding tasks. HoLoRA’s performance on other types of tasks, such as generative or multimodal tasks, remains untested, potentially limiting its applicability.
3. While HoLoRA ensures strict orthogonality, the paper does not explore in detail how this constraint affects feature representation, model stability, or training dynamics. A deeper analysis could provide more insight into HoLoRA’s impact.
4. While HoLoRA does not introduce new hyperparameters, factors like low-rank parameter r may still affect results. The lack of sensitivity analysis for various parameter configurations leaves questions about its robustness and tuning flexibility.
5. While the paper claims compatibility with other LoRA variants, empirical tests on these combinations are absent, leaving uncertainty about potential optimization challenges or performance trade-offs when integrating with other methods.
6. The paper claims that HoLoRA has advantages over other LoRA variants that also explore orthogonality, such as OLoRA. However, it avoids direct comparison with these methods, making it difficult to be convinced of HoLoRA’s performance and time efficiency benefits.

**Questions:**

No questions.

---

### Note · Authors · 2024-11-13

I have read and agree with the venue's withdrawal policy on behalf of myself and my co-authors.